# Improving Speech Translation by Fusing Speech and Text

**Wenbiao Yin[1], Zhicheng Liu[2],*, Chengqi Zhao[2], Tao Wang[2], Jian Tong[2], Rong Ye[2]**
[1]Department of Computer Science and Technology, Nanjing University
[2]ByteDance
wenbiaoyin@smail.nju.edu.cn
{liuzhicheng.lzc, zhaochengqi.d, wangtao.960826, tongjian, yerong}@bytedance.com

## Abstract

In speech translation, leveraging multimodal data to improve model performance and address limitations of individual modalities has shown significant effectiveness. In this paper, we harness the complementary strengths of speech and text to improve speech translation. However, speech and text are disparate modalities, we observe three aspects of modality gap that impede their integration in a speech translation model. To tackle these gaps, we propose **Fuse-Speech-Text** (**FuseST**), a cross-modal model which supports three distinct input modalities for translation: speech, text and fused speech-text. We leverage multiple techniques for cross-modal alignment and conduct a comprehensive analysis to assess its impact on speech translation, machine translation and fused speech-text translation. We evaluate FuseST on MuST-C, GigaST and newstest benchmark. Experiments show that the proposed FuseST achieves an average 34.0 BLEU on MuST-C En→De/Es/Fr (vs SOTA +1.1 BLEU). Further experiments demonstrate that FuseST does not degrade on MT task, as observed in previous works. Instead, it yields an average improvement of 3.2 BLEU over the pre-trained MT model. Code is available at https://github.com/WenbiaoYin/FuseST.

## 1 Introduction

Speech translation (ST) accepts speech signals as the input and outputs target translation. Speech translation can be broadly categorized into cascade system and end-to-end speech translation (E2E ST). Cascade system (Sperber et al., 2017; Zhang et al., 2019; Lam et al., 2021) usually combines automatic speech recognition (ASR) and machine translation (MT). The MT subsystem uses ASR transcripts as input, which provide clear expression but may contain errors stemming from ASR. While E2E ST (Tang et al., 2021a; Fang et al., 2022; Ye et al.,

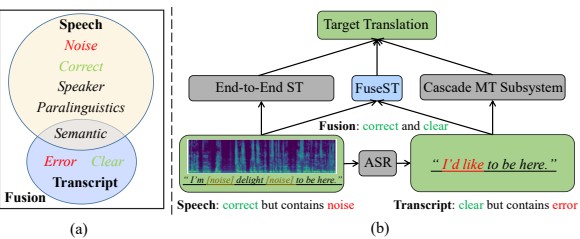

Figure 1: **Left**: Data distributions of speech and transcript. **Right**: The pipeline of our proposed FuseST.

2022) can directly map speech signals to the target translation, thus avoiding the problem of error propagation.

We observe a noticeable modality distribution difference between speech and ASR transcript as shown in Figure 1a. Speech signals contain abundant information, such as paralinguistics and speaker characteristics, but they are harder to model and more susceptible to noise. On the other hand, ASR transcripts are clear but may contain errors. For example, as shown in Figure 1b, the speech signal conveys the message "I'm delight to be here", but the speech signal contains numerous blank segments and background noise. E2E ST may encounter challenges in accurately extracting information directly from speech, particularly in the presence of noise, without compromising translation quality. Meanwhile, the ASR model mistakenly transcribes the speech as "I'd like to be here". The MT subsystem may proceed with translation without awareness of errors in "I'd like".

Inspired by these findings, we propose a model that fuses speech and ASR transcript (fused speech-text) as input to leverage their complementary strengths to improve speech translation. As shown in Figure 1b, our model supports three distinct modalities of input for translation: speech, text and fused speech-text.

However, speech and text are disparate modalities, we observe three aspects of modality gap that impede their integration in a speech transla-

---

*Corresponding author

tion model: **1.** The speech representation is in continuous space, while the text representation is in discrete space. **2.** When using speech and golden transcript as input, the model relies heavily on the golden transcript information and neglects the speech information. This behavior is because the golden transcript is more straightforward and contains precise semantic information, adequate for achieving high translation quality. The model will learn a shortcut to rely solely on the golden transcript. In contrast, speech representations obtained through the pre-trained model are harder to model, they include four types of information (Chen et al., 2022): content, semantics, paralinguistics and speaker characteristics. However, when using speech and ASR transcript as input, solely relying on textual information is insufficient to achieve good translation quality. We expect the model to incorporate more speech information. **3.** Diverse modality inputs lead to distinct hidden states in the encoder and distinct distributions in the decoder.

We propose several methods to bridge the modality gap to better integrate speech and text. **1.** We explore mapping continuous speech representation to a discrete space using a codebook to align with the text representation. **2.** We adopt the prompt tags implicitly guide the model to utilize more speech information when the transcript is inaccurate. We further explore how to explicitly guide the model to fuse speech and text. Meanwhile, we apply Cross-modal Contrastive Learning (CCL, Sohn, 2016) to reduce the gap between the model semantic of speech and its corresponding transcript. **3.** We adopt Cross-Attentive Regularization (CAR, Tang et al., 2021a) to align the states of the encoder and Cross-Modal Regularization (CMR) to align the distribution of the decoder.

Our contributions are summarized as follows:

- We propose a model that fuses speech and text to improve speech translation, which supports three distinct input modalities for translation: speech, text and fused speech-text.

- To fuse speech and text as input and leverage their complementary strengths, we conduct a comprehensive analysis of the modality gap between speech and text. We propose targeted improvements to bridge the modality gap between speech and text.

- Our experiments show that our model achieves an average 34.0 BLEU on MuST-

C En→De/Es/Fr (vs SOTA +1.1 BLEU) and achieves an average improvement of 3.2 BLEU over the pre-trained MT model on MuST-C.

## 2 Related Work

**Cascade ST** Cascade ST, achieved by concatenating ASR and MT components, has been extensively employed in commercial speech translation systems. However, cascade ST is vulnerable to challenges such as error propagation and high latency. To overcome the error propagation, ( Bertoldi and Federico, 2005; Beck et al., 2019; Sperber et al., 2019) proposed to feed the MT system with ASR data structures; ( Peitz et al., 2012; Cheng et al., 2019; Di Gangi et al., 2019a) proposed to make MT robust to ASR errors, for instance by training it on parallel data incorporating factual or emulated ASR errors.

**End-to-End ST** To overcome the error propagation and high latency in the cascade ST systems, (Bérard et al., 2016; Duong et al., 2016) proposed an end-to-end architecture for speech translation, which has attracted extensive attention ( Vila et al., 2018; Salesky et al., 2019; Gangi et al., 2019; Inaguma et al., 2021; Zhao et al., 2021). However, it is difficult to train an end-to-end speech translation model directly, primarily due to the inherent variability and complexity of speech signals and the scarcity of high-quality speech-translation datasets. Some training methods like pretraining ( Weiss et al., 2017; Bérard et al., 2018; Bansal et al., 2019; Wang et al., 2020a; Tang et al., 2021b), multi-task learning( Le et al., 2020; Vydana et al., 2021; Ye et al., 2021; Tang et al., 2022), data augmentation ( Park et al., 2019; Jia et al., 2019; Bahar et al., 2019; Pino et al., 2020), meta-learning ( Indurthi et al., 2020), contrastive learning ( Li et al., 2021; Ye et al., 2022), knowledge distillation ( Liu et al., 2019; Tang et al., 2021a) and curriculum learning ( Kano et al., 2017; Wang et al., 2020b), are proved to be effective.

## 3 Methods

### 3.1 Problem Formulation

The speech translation corpus is usually comprised of triples that include speech, transcript and target translation, which can be denoted as $D = (s, x, y)$. Here, $s$ is an audio sequence, $x$ is the corresponding transcript and $y$ is the corresponding target translation. Our model supports three distinct input modal-

ities for translation: speech (ST: $s \rightarrow y$), text (MT: $x \rightarrow y$) and fused speech-text (FT: $s + x \rightarrow y$).

## 3.2 Model Framework

As shown in Figure 2, our model consists of four sub-modules: Speech Encoder, Speech-Text Fusion Module, Transformer Encoder and Transformer Decoder.

**Speech Encoder** The speech encoder($S\text{-}Enc$) consists of Wav2vec2.0 (Baevski et al., 2020b) and two additional convolutional layers. The input is a raw waveform signal sampled at 16kHz, and Wav2vec2.0 is used to extract low-level speech representations from it. The two additional convolutional layers with stride 2 to shrink the speech length by a factor of 4. A greater degree of downsampling would have led to information loss, while a lesser degree of downsampling could have resulted in modal misalignment and compromised performance. Denote $a = S\text{-}Enc(s)$ as the speech representation.

To reduce the number of parameters and facilitate knowledge transfer, we share the *Transformer encoder* and *Transformer decoder* for ST, MT and FT.

**Transformer Encoder and Transformer Decoder** The *Transformer encoder* and *Transformer decoder* are composed of $N_e$ transformer encoder layers and $N_d$ transformer decoder layers, respectively, with the same configuration as the original implementation (Vaswani et al., 2017). We fisrt pre-train the model on external MT data and then optimize the whole model by minimizing the final loss. For the MT task, the input of the *Transformer encoder* is the embedding of transcript $e = Emb(x)$. For the ST task, the input is the audio output representation of the speech encoder $a = S\text{-}Enc(s)$. For the FT task, the input is the fused speech-text representation $f$(see details in Section 3.3). The *Transformer encoder* further extracts the high-level semantic hidden representations and facilitates knowledge sharing across the three modalities. The *Transformer decoder* generates corresponding target translation for ST, MT and FT. Besides, we train our model with auxiliary ASR task to improve translation performance. The training losses of ST, MT, FT and ASR are as follows:

$$\mathcal{L}_{ST} = -\sum_n \log P(y_n|s_n) \quad (1)$$

$$\mathcal{L}_{MT} = -\sum_n \log P(y_n|x_n) \quad (2)$$

$$\mathcal{L}_{FT} = -\sum_n \log P(y_n|s_n, x_n) \quad (3)$$

$$\mathcal{L}_{ASR} = -\sum_n \log P(x_n|s_n) \quad (4)$$

$\mathcal{L}_{ST}$, $\mathcal{L}_{MT}$, $\mathcal{L}_{FT}$ and $\mathcal{L}_{ASR}$ are cross-entropy losses on <speech, target>, <transcript, target>, <speech, transcript, target> and <speech, transcript>, respectively.

## 3.3 Fusing Speech and Text

To leverage the complementary strengths between speech and text, we propose the **Fuse-S**peech-**T**ext(**FuseST**) method. We first introduce FuseST in this section and later show how to bridge the modality gap between speech and text.

Here, we utilize an open-source ASR model to construct a dataset of <speech, ASR transcript, target> pair from the original dataset of <speech, golden transcript, target> pair. Given a speech-transcript-target pair $(s, x, y)$, the transcript $x$ could be an ASR transcript or golden transcript, and the fused speech-text representation $f$ is defined as:

$$f = Concat(P^s, S\text{-}Enc(s), P^t, P^f, Emb(x)) \quad (5)$$

where $P^s$ and $P^t$ are prompt tags to identify whether the input modal is speech or text, and $P^f$ is a prompt tag (<golden>/<asr>) to indicate whether the transcript is manually annotated or generated through ASR system. The prompt tags serve as implicit guidelines for the model, prompting it to harness a higher degree of speech information when the transcript is inaccurate, while inducing a decreased reliance on speech information when the transcript is deemed accurate.

## 3.4 Align Speech and Text with FuseST

As we analyzed in Section 1, we observe three aspects of the modality gap between speech and text. We propose several methods to bridge the modality gap between speech and text. In addition to the implicit guidance mentioned above, we introduce three additional methods in this section: Cross-modal Contrastive Learning, Cross-Attentive Regularization and Cross-Modal Regularization.

**Cross-modal Contrastive Learning** Given a positive example of speech-transcript $(s, x)$ pair, we

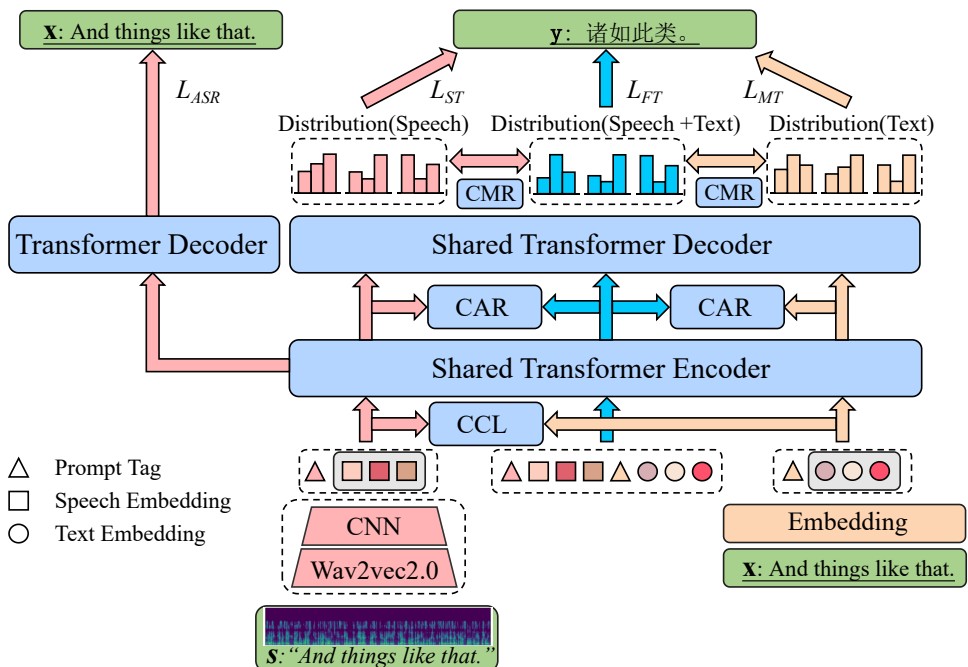

Figure 2: Overview of our proposed FuseST. FT denotes fused speech-text input for translation; CCL denotes Cross-modal Contrastive Learning; CAR denotes Cross-Attentive Regularization; CMR denotes Cross-Modal Regularization.

can get speech representation $a = S\text{-}Enc(s)$, transcript representation $e = Emb(x)$. We randomly pick a set of $B-1$ transcripts $\{e_i^-\}_{i=1}^{B-1}$ from the same batch as negative examples. For speech representation $a$ and transcript representation $e$, we first average them in terms of the time dimension and apply the multi-class N-pair contrastive loss (Sohn, 2016):

$$u = \text{MeanPool}(a) \qquad (6)$$

$$v = \text{MeanPool}(e) \qquad (7)$$

$$\mathcal{L}_{CCL} = -\sum_{s,x} \log \frac{\exp(sim(u,v)/\tau)}{\sum_{e_j \in \mathcal{A}} \exp(sim(u,v(e_j))/\tau)} \qquad (8)$$

where $\mathcal{A} = \{e\} \cup \{e_i^-\}_{i=1}^{B-1}$, $\tau$ is the temperature hyper-parameter and $sim$ is the cosine similarity function.

**Cross-Attentive Regularization** The cross-attentive regularization (Tang et al., 2021a) (CAR) can increase the similarity among distinct modalities. The essence of CAR is to use a similarity matrix to project a tensor sequence onto a space of equivalent length as another tensor sequence, then compute $L2$ loss between the two sequences. Here, we utilize the CAR to compute losses separately between speech representation $a$ and fused speech-text representation $f$ and

between text representation $e$ and fused speech-text representation $f$. The CAR loss is defined as:

$$\mathcal{L}_{CAR} = \mathcal{L}_{CAR}(a,f) + \mathcal{L}_{CAR}(e,f) \qquad (9)$$

**Cross-Modal Regularization** To bridge the modality gap in inference, we endeavor to optimize the congruity of the ultimate distributions of ST, MT and FT:

$$P(y|s) = P(y|s,x) = P(y|x) \qquad (10)$$

The ST task is more difficult than the MT task since the speech signals are harder to model and more susceptible to noise. Previous works (Liu et al., 2019; Gaido et al., 2020; Tang et al., 2021a) utilize knowledge distillation to facilitate knowledge acquisition by an ST model from a well-trained MT model. However, in our work, the transcript may contain errors; the accuracy of the FT task is usually much higher than the corresponding ST and MT task. We designate the FT task as the teacher while ST and MT tasks as the student, minimizing the loss between the student and teacher outputs. The KD loss is defined as:

$$\mathcal{L}_{KD} = \mathcal{L}_{KD}(a,f) + \mathcal{L}_{KD}(e,f) \qquad (11)$$

Furthermore, we minimize the Jensen-Shannon Divergence (Lin, 1991) (JSD) loss between the three output distributions, which is:

$$\mathcal{L}_{JSD} = \sum_n (JSD\{p(y_n|a_n)||p(y_n|,f_n)\}$$
$$+ JSD\{p(y_n|e_n)||p(y_n|f_n)\}) \quad (12)$$

The final training objective is as follows:

$$\mathcal{L} = \alpha\mathcal{L}_{ST} + \alpha\mathcal{L}_{MT} + (1-\alpha)\mathcal{L}_{KD} + \mathcal{L}_{FT}$$
$$+ \mathcal{L}_{ASR} + \mathcal{L}_{CCL} + \beta\mathcal{L}_{CAR} + \mathcal{L}_{JSD} \quad (13)$$

where $\alpha$ and $\beta$ are predefined hyper-parameters.

### 3.5 Inference

Our model supports three distinct modalities for translation: speech, text and fused speech-text. For FT task, given an audio sequence $s$, we use an open-source ASR model to get its corresponding transcript $x$, then fuse the audio sequence $s$ and its corresponding transcript $x$ as input and output target translation.

## 4 Experiments

### 4.1 Experimental Setups

**ST Datasets** We conduct experiments on MuST-C[1] (Di Gangi et al., 2019b) and GigaST[2] (Ye et al.). **MuST-C** is a multilingual speech translation dataset that contains translations from English to 8 languages. MuST-C contains several hundred hours of audio recordings from English TED Talks; we conduct experiments on MuST-C En→De/Es/Fr. We use the dev set for development and the tst-COMMON set for test. **GigaST** is a large-scale pseudo speech translation dataset created by translating the text in GigaSpeech. We conduct experiments on GigaST En→Zh, and test on in-house cgtn, zhiyuan and aiconf datasets. We utilize an open-source ASR model (whisper base.en [3]) to construct a dataset of <speech, ASR transcript, target> pair from original ST dataset. The Word Error Rate (WER) of our constructed datasets are shown in the last column of Table 1.

**MT Datasets** Our model allows us to use the external MT dataset for further training. We introduce external WMT datasets for En→De/Es/Fr and in-house MT dataset for En→Zh. The detailed statistics of all datasets are shown in Table 1.

| | ST | | MT Train | | MT Test | WER |
|---|---|---|---|---|---|---|
| | #hours | #sents | name | #sents | name | |
| MuST-C (Medium Resources) | | | | | | |
| En→De | 408 | 234K | WMT17 | 4.5M | newstest2014 | 22.65 |
| En→Es | 504 | 270K | WMT17 | 4.1M | newstest2013 | 22.60 |
| En→Fr | 492 | 280K | WMT17 | 5.5M | newstest2014 | 22.08 |
| GigaST (High resources) | | | | | | |
| En→Zh | 9,780 | 7,650K | in-house | 105M | newstest2019/2020 | 21.98 |

Table 1: Statistics of all datasets.

**Model Configuration** For the speech encoder, we use Wav2vec2.0[4] following the base configuration, which is only pre-trained on Librispeech (Panayotov et al., 2015) without any finetuning. Two layers of CNNs after the Wav2vec2.0 with kernel size 5, stride size 2, padding 2 and hidden dimension 1024. The transformer encoder and decoder follow the base configuration, with hidden size $h^d = 512$, 8 attention heads and 2048 FFN hidden states. We use $N_e = 6$ transformer encoder layers and $N_d = 6$ transformer decoder layers.

**Experiment Details** We first pre-train our model on the external MT dataset; the learning rate is 5e-4. We then optimize our model by minimizing the final loss; the learning rate is 6e-5. We use the raw 16kHZ speech as input and jointly tokenize the bilingual text using SentencePiece (Kudo and Richardson, 2018). We use an Adam optimizer with $\beta_1 = 0.9$, $\beta_2 = 0.98$ and 20k warm-up updates. The dropout is set to 0.15 and the value of label smoothing is set to 0.1. For the training loss, we set weight of $\mathcal{L}_{ST}$ and $\mathcal{L}_{MT}$ $\alpha = 0.8$, contrastive temperature $\tau = 0.02$ and weight of $\mathcal{L}_{CAR}$ $\beta = 0.02$. We use sacreBLEU[5] (Post, 2018) to evaluate case-sensitive detokenized BLEU.

### 4.2 Baseline Systems

We compare our method with cascade models and end-to-end baseline models including: Espnet (Inaguma et al., 2021), W2V2-Transformer (Fang et al., 2022), Ye et al., 2021, Xu et al., 2021, MTL (Tang et al., 2021b), FAT-ST (Zheng et al., 2021), JT-S-MT (Tang et al., 2021a), Chimera (Han et al., 2021), XSTNet (Ye et al., 2021), SATE (Xu et al., 2021), STEMM (Fang et al., 2022), TaskAware (Indurthi et al., 2021), STPT (Tang et al., 2022), ConST (Ye et al., 2022).

Besides, we implement several methods for fusing speech and text modalities. The only difference

[1] We use v1.0 https://ict.fbk.eu/must-c/
[2] https://st-benchmark.github.io/resources/GigaST
[3] https://huggingface.co/openai/whisper-base.en

[4] https://dl.fbaipublicfiles.com/fairseq/wav2vec/wav2vec_small.pt
[5] https://github.com/mjpost/sacrebleu, sacreBLEU signature: nrefs:1 | bs:1000 | seed:12345 | case:mixed | eff:no | tok:13a | smooth:exp | version:2.0.0

| Models | External Data | | | | BLEU | | | |
|---|---|---|---|---|---|---|---|---|
| | Speech | Text | ASR | MT | De | Es | Fr | Avg. |
| **Cascade Model** | | | | | | | | |
| Espnet (Inaguma et al., 2021) | - | - | - | - | 23.6 | - | 33.8 | - |
| W2V2-Transformer (Fang et al., 2022) | ✓ | - | - | ✓ | 26.9 | 30.0 | 36.6 | 31.2 |
| (Ye et al., 2021) | - | - | ✓ | ✓ | 25.2 | - | 34.9 | - |
| (Xu et al., 2021) | - | - | ✓ | ✓ | 28.1 | - | - | - |
| **End-to-End Model** | | | | | | | | |
| MTL (Tang et al., 2021b) | - | - | - | ✓ | 23.9 | 28.6 | 33.1 | 28.5 |
| FAT-ST (Zheng et al., 2021) | ✓ | ✓ | ✓ | ✓ | 25.5 | 30.8 | - | - |
| JT-S-MT (Tang et al., 2021a) | - | - | - | ✓ | 26.8 | 31.0 | 37.4 | 31.7 |
| Chimera (Han et al., 2021) | ✓ | - | - | ✓ | $27.1^{\dagger}$ | 30.6 | 35.6 | 31.1 |
| XSTNet (Ye et al., 2021) | ✓ | - | - | ✓ | 27.1 | 30.8 | 38.0 | 32.0 |
| SATE (Xu et al., 2021) | - | - | ✓ | ✓ | $28.1^{\dagger}$ | - | - | - |
| STEMM (Fang et al., 2022) | ✓ | - | - | ✓ | 28.7 | 31.0 | 37.4 | 32.4 |
| TaskAware (Indurthi et al., 2021) | - | - | ✓ | ✓ | 28.9 | - | - | - |
| STPT (Tang et al., 2022) | ✓ | ✓ | ✓ | ✓ | - | 33.1 | **39.7** | - |
| ConST (Ye et al., 2022) | ✓ | - | - | ✓ | 28.3 | 32.0 | 38.3 | 32.9 |
| **FuseST-ST** | ✓ | - | - | ✓ | 27.7 | 32.4 | 37.2 | 32.4 |
| **FuseST-FT** | ✓ | - | - | ✓ | **29.2** | **33.9** | 38.9 | **34.0** |

Table 2: Case-sensitive detokenized BLEU scores on MuST-C tst-COMMON set. "Speech" denotes unlabeled audio data, "Text" denotes unlabeled text data, *e.g.* Europarl V7 (Koehn, 2005), CC25 (Liu et al., 2020), † use 40M OpenSubtitles (Lison and Tiedemann, 2016) as external MT data.

| Models | De | | | Es | | | Fr | | | Avg. |
|---|---|---|---|---|---|---|---|---|---|---|
| | newstest | asr | golden | newstest | asr | golden | newstest | asr | golden | |
| Pre-trained 6E6D MT$^{\sharp}$ | 23.7 | 24.3 | 30.4 | 31.8 | 29.5 | 34.2 | 34.7 | 32.2 | 39.9 | 31.2 |
| Pre-trained 24E6D MT$^{\sharp}$ | **27.5** | 25.7 | 31.9 | **34.8** | 32.0 | 37.0 | **38.1** | 34.2 | 41.5 | 33.6 |
| STEMM (Fang et al., 2022) | - | - | 31.5 | - | - | - | - | - | - | - |
| **FuseST-MT** | 25.0 | **28.6** | **34.2** | 32.9 | **33.5** | **37.7** | 35.9 | **37.7** | **44.1** | **34.4** |

Table 3: Case-sensitive detokenized BLEU scores on MuST-C tst-COMMON set and newstest. "asr" denotes use the ASR transcript as input, "golden" denotes use the golden transcript as input. ♯ are trained on the same external MT data and MuST-C <transcript, target> pair data (including our constructed data).

between our approach and others is the specific method for fusing speech and text. SA-CTR: our implementation involved drawing inspiration from the image and text fusion techniques employed in Li et al.'s (2022) to propose a method for fusing speech and text. We utilize the method from Baevski et al. (2020a) to map the speech representation $e$ onto a discrete space using a codebook; then we concatenate discrete speech representation and text representation as ours. Codebook-Gumbel-Softmax: the Gumbel-Softmax quantization computes logits representing the codebook vectors; Codebook-K-means: K-means vector quantization computes the distance to all codeword vector and chooses the closest.

## 4.3 Main Results

**Comparison with End-to-End Baselines** As shown in Table 2, we compare our model with several strong end-to-end baselines. Many existing works rely on additional auxiliary data for better performance, *e.g.* large-scale MT data and unlabeled audio data. In the table, we provide a summary of the auxiliary data employed by these baselines, with a ✓ denoting its usage in the corresponding column. Our E2E FuseST-ST achieves comparable results with the previous best models. When fusing speech and ASR transcript as input, our FuseST-FT outperforms SOTA by an average 1.1 BLEU on MuST-C, demonstrating the superiority of our approach.

**Comparison with Cascade Baselines** We compare our model with several strong cascade systems. W2V2-Transformer, Ye et al. (2021) and Xu et al. (2021) provided three strong cascade systems trained using MuST-C and external ASR and MT data. As shown in Table 2, our E2E FuseST-ST achieves comparable results with these strong cascade models, while our FuseST-FT significantly outperforms these strong cascade models.

| Models | newstest2019 | newstest2020 | in-house cgtn | | | in-house zhiyuan | | | in-house aiconf | | |
|---|---|---|---|---|---|---|---|---|---|---|---|
| | | | asr | speech | fused | asr | speech | fused | asr | speech | fused |
| Pre-trained 6E6D MT♯ | 36.7 | 43.0 | 29.7 | - | - | 28.2 | - | - | 30.5 | - | - |
| Pre-trained 24E6D MT♯ | **40.1** | **46.6** | **32.6** | - | - | **29.5** | - | - | **33.6** | - | - |
| SA-CTR (Li et al., 2022) | 36.8 | 43.1 | 31.0 | 31.8 | 31.0 | 28.3 | 29.4 | 29.1 | 31.6 | 32.4 | 32.1 |
| Codebook-Gumbel-Softmax♭ | 36.8 | 43.0 | 30.9 | 30.0 | 31.7 | 28.1 | 28.4 | 28.9 | 31.7 | 31.6 | 32.5 |
| Codebook-K-means♭ | 36.9 | 43.2 | 31.0 | 31.3 | 31.8 | 28.2 | 29.3 | 29.0 | 31.5 | 31.7 | 32.3 |
| Align-Mask♮ | 38.3 | 43.9 | 30.8 | 32.1 | 31.7 | 28.4 | 29.8 | 29.4 | 32.1 | 33.1 | 33.0 |
| **FuseST** | 38.3 | 44.2 | 31.6 | **33.3** | **33.9** | 28.4 | **30.2** | **30.5** | 32.4 | **34.0** | **34.5** |

Table 4: Case-sensitive detokenized BLEU scores on En→Zh test sets. ♯ are training on the same MT data and GigaST <transcript, target> pair data (including our constructed data); ♭ are using the method from (Baevski et al., 2020a) to map the speech representation onto a discrete space using a codebook; ♮ is explicit guidance fusion(see detail in Section 5.6(6)). "asr" denotes use the ASR transcript as input, "fused" denotes use ASR transcript and speech as input. Hubert large is used to extract speech features on the GigaST En→Zh.

| Task | MT | MT | ST | FT |
|---|---|---|---|---|
| Config. | newstest | asr | speech | fused |
| FuseST | 25.0 | 28.6 | 27.7 | 29.2 |
| $-\mathcal{L}_{ASR}$ | 24.9 | 28.5 | 27.4 | 28.9 |
| $-\mathcal{L}_{ASR} - \mathcal{L}_{KD} - \mathcal{L}_{JSD}$ | 25.2 | 28.0 | 27.1 | 28.8 |
| $-\mathcal{L}_{ASR} - \mathcal{L}_{KD} - \mathcal{L}_{JSD} - \mathcal{L}_{CCL} - \mathcal{L}_{CAR}$ | 24.8 | 27.6 | 26.4 | 28.5 |

Table 5: BLEU scores on MuST-C En→De tst-COMMON set and newstest set by removing individual losses.

| Task | MT | MT | ST | FT |
|---|---|---|---|---|
| Model | newstest | asr | speech | fused |
| Wav2vec2.0 | 25.0 | 28.6 | 27.7 | 29.2 |
| HuBERT Large | 25.0 | 28.4 | 29.3 | 29.7 |
| HuBERT Extra Large | 25.1 | 28.4 | 30.0 | 30.3 |

Table 6: The impact of speech pre-trained model on MuST-C En→De tst-COMMON set and newstest set.

## 5 Analysis

### 5.1 Is FuseST better than other fusion methods?

We compare our model with different fusion methods, such as SA-CTR and STEMM. SA-CTR proposed a selective attention and gated fusion mechanism to fuse two different modalities; STEMM proposed the speech-text manifold mixup to mix up the representation sequences of different modalities. Our model achieves better results than theirs by utilizing a prompt-based approach.

### 5.2 Should speech representation be discrete or continuous?

As Section 1 mentions, speech input representation is in continuous space, while text input representation is in discrete space. We utilize the method from Baevski et al. (2020a) to map the speech representation onto a discrete space using a codebook. However, Codebook-Gumbel-Softmax and Codebook-K-means led to a decrease in BLEU score (shown in Table 4). Our conjecture here is that the existing speech pre-trained models extract continuous features that undergo a mapping process to a discrete space, resulting in the loss of audio information and a subsequent reduction in the BLEU score. Nonetheless, if the speech pre-trained models can extract high-quality discrete features, it is plausible that such discrete features could enhance performance.

**Comparison with MT Baselines** We fisrt pre-train our model on external MT data and then jointly train on multiple tasks. Previous work has encountered catastrophic forgetting problems on MT task during joint training (Fang et al., 2022), which significantly degrades performance on MT tasks. We evaluate our model on the MT task and show the result in Table 3 and Table 4. Our model achieves significant improvement on the MT task instead of a decline in performance. Our model even outperforms pre-trained 24E6D (24 transformer encoder layers and 6 transformer decoder layers) MT on MuST-C. However, with the increase of training data, our model performs lower than pre-trained 24E6D MT when using ASR transcript as input (En→Zh). Nevertheless, our model can still outperform pre-trained 24E6D MT when fusing speech and text as input on En→Zh.

### 4.4 Ablation Study

As shown in Equation 13, our training objective contains eight terms. In addition to the cross-entropy objective $\mathcal{L}_{ST}, \mathcal{L}_{MT}, \mathcal{L}_{FT}$, we investigate the effects of the other auxiliary training objectives. By gradually removing each loss, Table 5 shows the improvements brought by each auxiliary training objective.

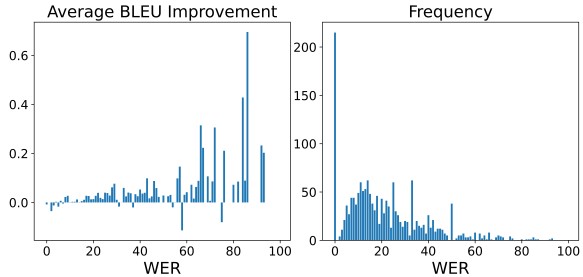

Figure 3: **Left**: The average BLEU improvement for ASR transcript resulting from using fuse-speech-text on En→Zh aiconf testset, under various conditions of WER. **Right**: The frequency distribution of WER on En→Zh aiconf testset.

| Models | | |
|---|---|---|
| | | CASE 1 |
| Ref. | **src**: | I'm extraordinarily delight to be here. |
| | **asr**: | I'm extraordinary. I'd like to be here. |
| | **tgt**: | 我非常高兴来到这里。 |
| FuseST-MT | **tgt** | 我非同寻常,我想在这里。 |
| FuseST-ST | **tgt** | 我非常高兴来到这里。 |
| FuseST-FT | **tgt** | 我非常高兴来到这里。 |
| | | CASE 2 |
| Ref. | **src**: | So you could think of the viruses like, they are people. |
| | **asr**: | So you could think of the viruses like their people. |
| | **tgt**: | 所以你可以把病毒想象成人。 |
| FuseST-MT | **tgt** | 所以你可以像他们的人民一样看待病毒。 |
| FuseST-ST | **tgt** | 所以你可以把病毒想象成他们的人。 |
| FuseST-FT | **tgt** | 所以你可以把病毒想象成人。 |

Table 7: Case study: cases are generated from FuseST-MT, FuseST-ST and FuseST-FT on En→Zh aiconf testset. The red underlined text indicates inaccurate translations, and the blue strikethrough indicates missing translation.

## 5.3 With the increasing advancements of speech pre-trained models, does the fusion of speech and text remain effective?

Here, we report the results of Wav2vec2.0, HuBERT Large[6] and HuBERT Extra Large[7], which are widely used in speech translation. As shown in Tabel 6, as the strength of speech pre-trained models increases, the performance of the models on ST and FT improves. Nevertheless, as speech pre-trained models undergo further advancements, the marginal gains in speech translation resulting from text fusion have shown a diminishing trend. This phenomenon can be ascribed to the progressive refinement of speech representations, in contrast to the relatively inferior quality of our textual representations (WER ≈ 20). By incorporating stronger textual representations, the enhancement in speech translation through text would become more pronounced.

## 5.4 How does our fusing strategy perform on different levels of ASR transcript quality?

We conduct an empirical study to examine the efficacy of our approach in enhancing BLEU score through the fusion of speech information under different WER present in the ASR transcript. As shown in Figure 3, when the word error rate of ASR transcript is minimal ($0 \leq$ WER $< 5$), the fusion of speech information results in a slight decrease in BLEU score. This behavior is because the transcript is adequate for achieving high translation quality, and the fusion of speech information may introduce noise. As the WER increases, the ad-

[6]https://dl.fbaipublicfiles.com/hubert/hubert_large_ll60k.pt
[7]https://dl.fbaipublicfiles.com/hubert/hubert_xtralarge_ll60k.pt

vantages of integrating speech information become more pronounced.

## 5.5 Is our model robust to different ASR errors?

In the GigaST En→Zh experiment, we utilize an ASR model different from the one used to construct the training set to generate the ASR transcripts for the test (in-house cgtn, zhiyuan and aiconf). As shown in Table 4, using fused speech-text for translation still outperforms using ASR transcript. Our model demonstrates strong robustness to ASR errors under various distributions.

## 5.6 What is the comparative effectiveness between implicit guidance and explicit guidance in the fusion process?

In Section 3.3, we utilize prompt tags to implicitly guide the model to fuse speech and text. This section further explores how to explicitly guide the model to fuse speech and text. We propose an additional auxiliary task named Align-Mask: we first use the Montreal Forced Aligner(MFA, (McAuliffe et al., 2017)) toolkit to get word-level speech-transcript alignment pairs; then we random mask a consecutive sequence of 1 to 4 words in the text with a probability of 15%, we only preserve the speech segment corresponding to the masked text segment; then we predict the masked words. We aspire for the model to learn the correspondence between speech and text to fuse speech and text better. However, Align-Mask performs worse as shown in Table 4. Our hypothesis is that using MFA to align introduces errors, which affect the effectiveness of Align-Mask. In the future, we will further explore explicit guidance fusion without external tools.

## 6 Case Study

In this section, we present several cases generated by FuseST-MT, FuseST-ST and FuseST-FT. In the first case, the ASR model mistakenly transcribes "delight" as "I'd like" due to the highly similar pronunciations of these two words. FuseST-MT fails to generate the correct translation as a result of errors present in the ASR transcript. Meanwhile, FuseST-ST produces omissions, as modeling direct speech to target translation proves to be more challenging. Notably, only FuseST-FT translates the sentence correctly, leveraging the complementary strengths of speech and text. In the second case, the speech signal contains numerous background noises; the ASR model mistakenly transcribes the "they are" as "their", FuseST-MT and FuseST-ST are mistranslated, and only FuseST-FT translates correctly.

## 7 Conclusion

In this paper, we propose FuseST, a cross-modal model which supports three distinct input modalities for translation: speech, text and fused speech-text. We comprehensively analyze the modality gap between speech and text and, utilize multiple techniques to bridge the modality gap. We then fuse speech and text to improve speech translation. Experiments and analysis demonstrate the effectiveness of our proposed method.

## Limitations

This work improves speech translation by fusing speech and text, but the model is far from being achieved for industrialgrade implementations. Although the ChatGPT and Whisper models exhibit superior speech-to-text capabilities compared to our model, we maintain that fusing speech and text remains a viable approach in the era of large-scale models. There are two significant limitations in this study that could be addressed in future research. First, our model still relies on an ASR system to transcribe speech into text, which does not address the issue of high latency in the cascade system. Second, our model needs labeled data for training, especially the <speech, transcript, target> pair. Speech data is exceptionally scarce, and obtaining speech data for many languages around the world is particularly challenging.

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
