# OpenReview forum: "Improving Speech Translation by Fusing Speech and Text"
_EMNLP/2023/Conference — EMNLP 2023 Findings_

### Official Review · Reviewer_VGMC · 2023-07-24

**Soundness:** 3

**Excitement:**

3: Ambivalent: It has merits (e.g., it reports state-of-the-art results, the idea is nice), but there are key weaknesses (e.g., it describes incremental work), and it can significantly benefit from another round of revision. However, I won't object to accepting it if my co-reviewers champion it.

**Paper Topic And Main Contributions:**

This paper propose a framework, that take pre-trained representation for speech, text embedding (from either ASR transcript or groundtruth) and learn a joint representation between those by introduce different modality matching loss. Results show proposed method could improve speech translation on MuSTC (for some language pair) by more than  1 BLEU.

The contribution of the paper has 2 folds:
(1) Propose a framework to use fused information of speech and text representation to help speech translation.
(2) Tested many different modality matching function to help align audio and text embeddings.

**Reasons To Accept:**

The motivation is clear. For speech translation, speech contains some signal to help translation but if ASR doesn't capture it then error would propagate to the final translation quality.

**Reasons To Reject:**

(1) The proposed method is complicated. It combines pretrained audio, text modular with 8 different losses. This make it hard to conclude the improvement is came from proposed model or just introduce many hyper-parameters to tune.

(2) The comparison missed one direction of fuse speech and text representation for speech translation. For example, MAESTRO: Matched Speech Text Representations through Modality Matching (https://arxiv.org/abs/2204.03409). That basically learn a joint encoder can encode both audio and text. Can the author comment the pros and cons of this direction versus proposed method?

(3) Presentation is hard to follow. For example, it mentioned all encoder decoder share the same param in section 3.2, but why figure 2 mark some of param shared? There also lots of acronym in the paper, that make it hard to follow.

(4) It's hard to follow the comparison, since external modular been used. In table 2, when mark as external speech/text data been used, does it comparable with other cited paper? Maybe to make at least one data point that comparable, can the author provide their in-house cascade model, instead of cite other paper number?

**Reproducibility:**

3: Could reproduce the results with some difficulty. The settings of parameters are underspecified or subjectively determined; the training/evaluation data are not widely available.

**Reviewer Confidence:**

4: Quite sure. I tried to check the important points carefully. It's unlikely, though conceivable, that I missed something that should affect my ratings.

**Typos Grammar Style And Presentation Improvements:**

Please do not call it "FST". This is very confusing, because FST usually refer to finite state transducer for transcription task.

---

> ### Author Rebuttal · Authors · 2023-08-26
>
> Thanks for your careful and valuable comments. We will explain your concerns point by point.
> ### **Answer to "Reasons To Reject" :**
> **Q1: The proposed method is complicated. It combines pretrained audio, text modular with 8 different losses. This make it hard to conclude the improvement is came from proposed model or just introduce many hyper-parameters to tune.**
>
> **A1:** Our model incorporates 8 loss terms, comprising four fundamental tasks (ST, MT, FT, ASR) and four alignment losses. Previous methods primarily focused on improving the model's performance on the ST task, neglecting the efficacy of other translation tasks, resulting in fewer loss functions employed. In contrast, we aimed to enhance the performance of our model across all translation tasks (ST, MT, FT). Our model not only achieves a significant improvement of 1.1 BLEU in ST but also demonstrates a notable enhancement of 3.2 BLEU in MT. Meanwhile, the parameter size of our model is nearly consistent with the models compared in Table 2. We will provide explicit annotations for the parameter sizes of each model in the final version.
>
> Based on equation 13, it is evident that our loss function incorporates solely two hyperparameters. Importantly, the configurations of these hyperparameters mirror those employed in previous methods, and we have not adjusted their values. Consequently, we attribute the observed improvements in model performance primarily to the effectiveness of our proposed model rather than any fine-tuning of hyperparameters.
>
> **Q2:The comparison missed one direction of fuse speech and text representation for speech translation. For example, MAESTRO: Matched Speech Text Representations through Modality Matching (https://arxiv.org/abs/2204.03409). That basically learn a joint encoder can encode both audio and text. Can the author comment the pros and cons of this direction versus proposed method?**
>
> **A2:** Our paper contains a comparison with the direction of fuse speech and text representation for speech translation. For instance, Table 2  includes Chimera(https://aclanthology.org/2021.findings-acl.195.pdf), STEMM(https://aclanthology.org/2022.acl-long.486.pdf), ConST(https://aclanthology.org/2022.naacl-main.376.pdf).
>
> We have previously taken notice on the MAESTRO approach. However, we refrain from conducting a comparative analysis with MAESTRO for two primary reasons. Firstly, MAESTRO's experimentation was focused on CoVoST 2, and secondly, the absence of a code repository link in the corresponding paper hindered our ability to perform a comprehensive evaluation. Consequently, we could not establish a fair and equitable comparison with MAESTRO on the MuST-C dataset. Our approach differs from MAESTRO in the following aspects: our paper emphasizes integrating speech and text to enhance translation performance; to bridge the modality gap for a more seamless integration of speech and text, we propose FT(s + x → y) task and present several methods for cross-modal alignment.
>
> The pros and cons of our model compared to the previous method are:
>
> **Pros:**
>
> (a) We propose a model that fuses speech and text to improve speech translation, which supports three distinct input modalities for translation: speech, text and fused speech-text.
>
> (b) We propose several methods to bridge the modality gap to integrate speech and text better.
>
> (c) Our model not only achieves a significant improvement of 1.1 BLEU in ST but also demonstrates a notable enhancement of 3.2 BLEU in MT
>
> **Cons:**
>
> (a) Our model is more complex than the previous method. However, this increased complexity is deliberate, as it aligns with our objective of enhancing the effectiveness of ST, MT and FT.
>
> **Q3: Presentation is hard to follow. For example, it mentioned all encoder decoder share the same param in section 3.2, but why figure 2 mark some of param shared? There also lots of acronym in the paper, that make it hard to follow.**
>
> **A3:** In section 3.2 Line 189-191, "We share the Transformer encoder and Transformer decoder for ST, MT and FT." A shared transformer encoder and shared decoder (for ST, MT and FT); and a non-shared transformer decoder (for the ASR task only). In the final version, we will make a concerted effort to minimize the usage of acronyms in our paper and refine our writing accordingly. We appreciate your valuable suggestion.
>
> **Q4: It's hard to follow the comparison, since external modular been used. In table 2, when mark as external speech/text data been used, does it comparable with other cited paper? Maybe to make at least one data point that comparable, can the author provide their in-house cascade model, instead of cite other paper number?**
>
>
> **A4:** In Table 2, our comparisons with the cited methods are valid and based on comparable settings. We employ the same dataset as many referenced approaches, such as ConST and STEMM. How we compare our results with previous methods remains consistent, and you can verify this by referring to the experimental tables of ConST(https://aclanthology.org/2022.naacl-main.376.pdf) and STEMM(https://aclanthology.org/2022.acl-long.486.pdf).
>
> We have also included the in-house cascade model in Table 3 and Table 4, where "Pre-trained 6E6D MT" corresponds to the cascaded model with the same parameters. Our approach exhibits significant superiority over the in-house cascaded model with the same parameters.
>
> ### **Answer to "Typos Grammar Style And Presentation Improvements":**
>
> We sincerely apologize for the confusion caused by the abbreviation FST. Henceforth, we will proceed to modify the model's name accordingly. We appreciate your valuable suggestion.

---

### Official Review · Reviewer_dtrK · 2023-08-06

**Typos Grammar Style And Presentation Improvements:** 1) A small comment, but the acronym F…
**Soundness:** 4

**Excitement:**

3: Ambivalent: It has merits (e.g., it reports state-of-the-art results, the idea is nice), but there are key weaknesses (e.g., it describes incremental work), and it can significantly benefit from another round of revision. However, I won't object to accepting it if my co-reviewers champion it.

**Paper Topic And Main Contributions:**

This paper proposes a model called Fuse-Speech-Text (FST) for end-to-end speech translation. The pre-training involves several objectives for unimodal learning and cross-modal alignment. The FST model is jointly trained to optimize for speech-translation (ST), machine translation (MT), fused speech-text translation (FT) and ASR. The model consists of : a) a speech-encoder (Wav2vec2.0 + 2CNN layers); b) a shared transformer encoder and shared decoder (for ST, MT and FT); and c) a non-shared transformer decoder (for the ASR task only). Further, to encourage cross-modal alignment, the model uses : a) cross-modal contrastive learning; b) cross-attentive regularization that has been used to increase similarity amongst distinct modalities and c) cross-modal regularization using knowledge distillation, where the FT task is the teacher and the ST and MT tasks are the student. Experiments are conducted on the MuST-C and GigaST datasets for ST and WMT datasets for MT; and gains are observed on both tasks: +1.1 BLEU over baselines for former and +3.2 BLEU for MT after training with the FST objectives (the FST model is first pre-trained for the MT task).

**Reasons To Accept:**

A) Experiments and Code: The experimental section is thorough and the paper achieves good results with improvements on machine translation as well. The paper includes several baselines to compare against. The code is also released, good job with this!

B) The figures in the paper are intuitive, color-coded and easy-to-understand.

C) The model framework section (3.2) is clearly explained, and nicely builds up to Equation 13 which has 8 loss terms.

D) The paper is also well-scoped in that the aim is clearly defined as speech translation, and the way to improve it is by improving cross-modal alignment. The paper does not have more or less to offer than what it claims which is a great thing.

**Reasons To Reject:**

A) Writing: The paper might benefit with revisions in writing. For example, L377 mentions that the "only difference between our approach and others is the specific method for fusing speech and text". Accordingly, the related work section should talk about these works and discuss how the paper is different from them. In its current state, its mostly a list of papers and their citations. L096-101 are also unclear, since prompt tags are not introduced before this.

B) The overall framework optimizes for 8 objectives and it is unclear how each help and why they are included. Although the paper has a lot of content to offer, a more intuitive approach for the inclusion of each regularization term even at the cost of a few experiments, would be beneficial, since a major contribution of the paper is the model. For example, stating something like: "analysing previous model outputs or systems reveal certain limitations which can be overcome by these additional objectives ...".

The complexity of the framework and non-clarity on each objective makes me hesitant on the excitement front, but I find the study to be thorough.

**Reproducibility:**

5: Could easily reproduce the results.

**Reviewer Confidence:**

3: Pretty sure, but there's a chance I missed something. Although I have a good feel for this area in general, I did not carefully check the paper's details, e.g., the math, experimental design, or novelty.

---

> ### Author Rebuttal · Authors · 2023-08-26
>
> Thanks for your careful and valuable comments. We will explain your concerns point by point.
> ### **Answer to "Reasons To Reject" :**
> **(A)** Writing: Below, we present some previously insufficiently elaborated details:
> - **Further clarification for Line 377:** The only distinction between our approach and the subsequent methods, namely SA-CTR, Codebook-Gumbel-Softmax, and Codebook-K-means, lies in the manner of integrating speech and text for translation purposes (FT: s + x → y). However, other aspects such as speech feature extraction, text encoding, and training loss remain consistent with our methods. For specific details regarding the discrepancies, please refer to Lines 379-392.
> - **Our key distinctions from related works are as follows:** our paper emphasizes integrating speech and text to enhance translation performance; we first incorporate speech and transcript for translation(FT: s + x → y); we propose several methods to bridge the modality gap to better integrate speech and text.
> - **List of papers and their citations:** In Table 2, we compared our method with cascade models and end-to-end baseline models including: Espnet (Inaguma et al., 2021), W2V2-Transformer (Fang et al., 2022), MTL (Tang et al., 2021b),FAT-ST (Zheng et al., 2021), JT-S-MT (Tang et al., 2021a), Chimera (Han et al., 2021), XSTNet (Ye et al., 2021), XSTNet (Ye et al., 2021), SATE (Xu et al., 2021), STEMM (Fang et al., 2022), TaskAware (Indurthi et al., 2021), STPT (Tang et al., 2022), ConST (Ye et al., 2022a).
>
> In the forthcoming final version, an additional page of content will be included specifically to provide comprehensive elucidation on those aspects that were previously unclear.
>
>
>
>
> **(B)** Our model incorporates 8 loss terms, comprising four fundamental tasks (ST, MT, FT, ASR) and four alignment losses. Previous methods primarily focused on improving the model's performance on the ST task, neglecting the efficacy of other translation tasks, resulting in fewer loss functions employed. In contrast, we aimed to enhance the performance of our model across all translation tasks (ST, MT, FT). Our model not only achieves a significant improvement of 1.1 BLEU in ST but also demonstrates a notable enhancement of 3.2 BLEU in MT.
>
>  While we have conducted preliminary analyses on the impact of different loss functions on our model's performance through ablation study(Table 5), we have yet to extensively examine each loss function individually. By gradually removing each loss, Table 5 shows the improvements brought by each auxiliary training objective. Meanwhile, it has been demonstrated in previous studies that multi-task can enhance model performance, such as TaskAware (https://ieeexplore.ieee.org/document/9414703).
>
>
> ### **Answer to "Typos Grammar Style And Presentation Improvements":**
>  **Q1: A small comment, but the acronym FST can get confusing because many people associate it with finite-state transducer :)**
>
> **A1:**  We sincerely apologize for the confusion caused by the abbreviation FST. Henceforth, we will proceed to modify the model's name accordingly. We appreciate your valuable suggestion.
>
> **Q2: Equation 13 has 8 loss terms and is a lot to unpack. Although there is a lot of content in the paper, it might be nice to add some intuitions/analysis on which training objective or regularization technique provides for maximum benefits to bridge the cross-modal gap that motivates the construction of the FST model.**
>
> **A2:** As mentioned in the preceding " Answer to "Reasons To Reject" (B)", we have conducted preliminary analyses on the impact of different loss functions on our model's performance through ablation study(Table 5). Table 5 shows the improvements brought by each auxiliary training objective. In the final version, we plan to conduct experiments to assess the contribution of each loss function to alignment and provide detailed explanations in this regard.
>
>
> **Q3: L102-105 miss citations and don't explain these terms.**
>
> **A3:** We provide a detailed explanation and references for these terms in Section 3.4. We will add the missing information in the final version.
>
> Cross-Attentive Regularization (CAR, Tang et al., 2021a) can increase the similarity among distinct modalities.
>
> Cross-Modal Regularization (CMR): To bridge the modality gap in inference, we endeavor to optimize the congruity of the ultimate distributions of ST, MT and FT.

---

### Official Review · Reviewer_xQGj · 2023-08-07

**Soundness:** 4

**Excitement:**

3: Ambivalent: It has merits (e.g., it reports state-of-the-art results, the idea is nice), but there are key weaknesses (e.g., it describes incremental work), and it can significantly benefit from another round of revision. However, I won't object to accepting it if my co-reviewers champion it.

**Paper Topic And Main Contributions:**

The paper discussed the topic of speech and text joined training for speech translation model, aka fuse-speech-text training. The authors proposed hierarchical architecture to handle multi-modal input. First, a speech only and a text only encoder is applied separately on each modality to have the representations. Further, the representations are concatenated in the same sequence with a tag indicating the modality. A shared encoder is then applied on top of the joined sequence for training. Additionally, several losses are applied to the training, such as cross-model contrastive learning, cross attentive regularization and cross model regularization.

**Questions For The Authors:**

1. Line 243, how do you determine the ASR errors? Does P^f indicate the source of the text or the correctness of the text. The text from ASR can also be correct.
2. CTR is a bad abbreviation of cross-modal contrastive learning. It introduces confusion with the other two loss when reading the paper.
3. Figure 1 is too small to read.
4. Are the decoding two stages? From eq.5, the model needs text embedding to run the share encoder.

**Reasons To Accept:**

1. The strategy is simple and straightforward.
2. The numerical results show significant improvements over the baseline.

**Reasons To Reject:**

My major concern is the novelty of paper.  Similar ideas on architecture, contrastive learning and cross attention regularization have been proposed by several other papers, e.g. [1][2]. The design of the model is quite similar to [1] and [2] expect that the shared encoder takes concatenate input instead of separated input.

[1] Unified Speech-Text Pre-training for Speech Translation and Recognition
[2] WACO: Word-Aligned Contrastive Learning for Speech Translation

**Reproducibility:**

4: Could mostly reproduce the results, but there may be some variation because of sample variance or minor variations in their interpretation of the protocol or method.

**Reviewer Confidence:**

4: Quite sure. I tried to check the important points carefully. It's unlikely, though conceivable, that I missed something that should affect my ratings.

---

> ### Author Rebuttal · Authors · 2023-08-26
>
> Thanks for your careful and valuable comments. We will explain your concerns point by point.
> ### **Answer to "Reasons To Reject" :**
> Regarding the concerns about the novelty of our paper, the overall framework of most speech translation models maintains consistency due to its demonstrated effectiveness. This framework typically encompasses primary tasks (Speech Translation) and auxiliary tasks (Machine Translation, Automatic Speech Recognition, user-defined tasks etc.). Our approach aligns with this established framework; our paper emphasizes integrating speech and text to enhance translation performance. Furthermore, we aim to bridge the modality gap for a more seamless integration of speech and text. To achieve this, we propose a fused speech-text (FT: s + x → y) task and present several methods for cross-modal alignment. Through extensive experimentation, we validate the effectiveness of our approach and provide valuable insights into the fusion of speech and text. Notably, our model not only achieves a significant improvement of 1.1 BLEU in ST but also demonstrates a notable enhancement of 3.2 BLEU in MT.
>
> ### **Answer to "Questions For The Authors" :**
> **Q1: Line 243, how do you determine the ASR errors? Does P^f indicate the source of the text or the correctness of the text. The text from ASR can also be correct.**
>
> A1: We apologize for the confusion in our previous statement. Here, "P^f" represents the transcript, denoting whether it is manually annotated or generated through ASR.
>
> **Q2: CTR is a bad abbreviation of cross-modal contrastive learning. It introduces confusion with the other two losses when reading the paper.**
>
> A2. We have indeed considered the issue of the abbreviation for Cross-modal Contrastive Learning. Ultimately, we opted to maintain consistency with prior methods in our abbreviation. However, this decision may potentially mislead readers. We will rectify this issue in the final version. We appreciate your valuable suggestion.
>
> **Q3: Figure 1 is too small to read.**
>
> A3. We will revise Figure 1 to ensure its clarity and legibility in the final version. We appreciate your valuable suggestion.
>
> **Q4: Are the decoding two stages? From eq.5, the model needs text embedding to run the share encoder.**
>
> A4. No. Our model simultaneously utilizes speech and text as inputs; however, we only decode once.

---

### Meta-Review · Area_Chair_ghqk · 2023-09-14

**Recommendation:** 4

**Metareview:**

This paper proposes to improve speech translation by using a "fused" representation consisting of a combination of the original input speech and output produced by an ASR system. The reviewers found that the motivation for the approach was clear, with thorough experiments that supported the paper's claims (strong soundness scores). However the reviewers also all indicated that they are ambivalent about the excitement from this paper, in large part because of the complexity of the resulting system. There were also some issues with writing and clarity, but the authors largely addressed these in the rebuttal period.

---

### Decision · Program_Chairs · 2023-10-07

**Decision:**

Accept-Findings

**Comment:**

This paper proposes to improve speech translation by using a "fused" representation consisting of a combination of the original input speech and output produced by an ASR system. The reviewers found that the motivation for the approach was clear, with thorough experiments that supported the paper's claims (strong soundness scores). However the reviewers also all indicated that they are ambivalent about the excitement from this paper, in large part because of the complexity of the resulting system. There were also some issues with writing and clarity, but the authors largely addressed these in the rebuttal period.